# Exercise Training and Skeletal Muscle Antioxidant Enzymes: An Update

**DOI:** 10.3390/antiox12010039

**Published:** 2022-12-25

**Authors:** Scott K. Powers, Erica Goldstein, Matthew Schrager, Li Li Ji

**Affiliations:** 1Department of Health Sciences, Stetson University, Deland, FL 32723, USA; 2Department of Kinesiology, University of Minnesota, St Paul, MN 55455, USA

**Keywords:** endurance exercise, resistance exercise, high-intensity interval training, oxidative stress

## Abstract

The pivotal observation that muscular exercise is associated with oxidative stress in humans was first reported over 45 years ago. Soon after this landmark finding, it was discovered that contracting skeletal muscles produce oxygen radicals and other reactive species capable of oxidizing cellular biomolecules. Importantly, the failure to eliminate these oxidant molecules during exercise results in oxidation of cellular proteins and lipids. Fortuitously, muscle fibers and other cells contain endogenous antioxidant enzymes capable of eliminating oxidants. Moreover, it is now established that several modes of exercise training (e.g., resistance exercise and endurance exercise) increase the expression of numerous antioxidant enzymes that protect myocytes against exercise-induced oxidative damage. This review concisely summarizes the impact of endurance, high-intensity interval, and resistance exercise training on the activities of enzymatic antioxidants within skeletal muscles in humans and other mammals. We also discuss the evidence that exercise-induced up-regulation of cellular antioxidants reduces contraction-induced oxidative damage in skeletal muscles and has the potential to delay muscle fatigue and improve exercise performance. Finally, in hopes of stimulating further research, we also discuss gaps in our knowledge of exercise-induced changes in muscle antioxidant capacity.

## 1. Introduction 

The discovery that the protein erythrocuprein functions as an enzyme catalyzing the dismutation of superoxide radicals played a key role in advancing the field of free radical biology [1]. This antioxidant enzyme was labeled superoxide dismutase (SOD) and is located throughout human and animal tissues [1]. Indeed, the finding that SOD is widespread across tissues provided the first evidence that this enzyme likely plays a vital role in protecting cells against the damaging effects of the superoxide radical (O_2_^•−^) [1]. Subsequently, studies have identified additional antioxidant enzymes that work alongside of SOD to eliminate harmful free radicals and other reactive oxygen species (ROS) in cells; collectively, these enzymes form the intracellular antioxidant enzyme network. 

The discovery of SOD had a major impact on research in free radical biology. Similarly, the landmark observation that contracting skeletal muscles produce ROS launched the field of muscle redox biology [2]. Soon after this milestone finding, the first evidence that endurance exercise training increases the activities of select antioxidant enzymes in skeletal muscles (e.g., superoxide dismutase 1 (SOD1)) was reported [3]. This discovery led to additional studies investigating the impact of endurance exercise training on the activities of three key antioxidant enzymes: (1) superoxide dismutase; (2) glutathione peroxidase; and (3) catalase. These early findings have been reviewed in several reports [4,5,6]. Notably, in the past decade, additional studies using a variety of exercise training modalities have markedly expanded our understanding of exercise-induced changes in the antioxidant capacity of skeletal muscles in both animals and humans. Therefore, this is an appropriate time to summarize our current knowledge about exercise-induced changes in skeletal muscle antioxidant capacity. Specifically, this brief review discusses our present knowledge of the exercise training-induced changes in muscle antioxidant enzymes following endurance exercise training, high-intensity interval training, and resistance exercise training in both preclinical and human studies. By design, this review is concise, and we apologize to authors that are not cited in this report. We begin our review with a discussion of the sites of ROS production in contracting skeletal muscles. 

## 2. Intracellular Sources of ROS in Contracting Skeletal Muscles

The discovery that muscular contractions promote ROS production in skeletal muscles was first reported over 40 years ago [2], and this ground-breaking finding has been confirmed by numerous studies (reviewed in [7]). Indeed, it is established that O_2_^•−^ and nitric oxide (NO) are the primary free radical species produced by skeletal muscle and generation of both species is increased by muscular contractions (reviewed in [4,8]). The search for the intracellular sites of O_2_^•−^ and NO production in contracting skeletal muscles has spanned over 40 years. Clearly, NO production in muscle fibers occurs via nitric oxide synthases [9]. In contrast, debate continues about the primary intracellular sources of O_2_^•−^ production in contracting skeletal muscles [10]. This lack of consensus results from the fact that numerous sites of O_2_^•−^ production exist within skeletal muscles and it is technically challenging to identify the specific site(s) of O_2_^•−^ generation within contracting muscle fibers. Commonly proposed locations of O_2_^•−^ production within or near contracting muscle fibers include mitochondria, phospholipase A2 (PLA2), xanthine oxidase (XO), and nicotinamide adenine dinucleotide oxidases (NOX) (Figure 1). For details of the evidence supporting O_2_^•−^ production from each of these sites in contracting muscle the reader is referred to the following comprehensive reviews [11,12]. 

Although mitochondria can produce ROS at several sites, a growing consensus indicates that muscle contraction-induced increases in ROS production are not from mitochondrial origin and that a key source of ROS production within contracting muscles is NOX associated with the plasma membrane and the triads/transverse tubules [10,11,12]. Two isoforms of NOX exist in skeletal muscle; the NOX 2 isoform is located within the muscle membrane (sarcolemma) whereas the NOX4 isoform is found in both mitochondria and the sarcoplasmic reticulum as well as the capillary endothelium [13]. Detailed studies investigating the roles of NOX in muscle ROS production suggest that NOX2 is likely the predominant ROS producing system in contracting muscle fibers but crosstalk between NOX2, NOX4, and mitochondrial ROS production can also occur [10,12].

Although NOX isoforms are likely a key source of contraction-induced ROS, evidence also exists that PLA2 is involved in exercise-mediated increases in ROS production within muscle fibers [14,15]. Further, several lines of evidence also implicate XO in exercise-induced ROS production. Specifically, although XO is not expressed in skeletal muscle fibers, this enzyme is found within capillary endothelial cells surrounding muscle fibers and muscular contractions activate XO to produce O_2_^•−^ radicals outside the muscle fiber [16,17]. Moreover, NOX4 located within the capillary endothelial cells is also capable of producing ROS during muscular contractions [18]. Following O_2_^•−^ conversion to hydrogen peroxide (H_2_O_2_) via extracellular SOD, H_2_O_2_ can cross the sarcolemma to enter muscle fibers and elicit prooxidative reactions. 

In summary, evidence indicates that contracting skeletal muscles produce ROS at several locations within and outside (e.g., blood vessels) the muscle fiber. Growing evidence suggests that the primary source of ROS during muscular contraction is derived from NOX with the NOX2 isoform playing a key role in ROS production [10,11,12,19,20]. 

## 3. Cellular Antioxidant Enzymes

The term antioxidant has been defined in several ways. In the context of this review, antioxidants are defined as molecules that prevent or lower the rate of oxidation of a substrate [4]. Using this definition, nature has provided organisms with two layers of antioxidants to limit the oxidation of cellular molecules. One layer consists of small non-enzymatic molecules that defend against oxidation; examples include uric acid, glutathione (GSH), and dietary antioxidants such as vitamins E and C. Although many nutritional antioxidants can eliminate radicals via one-electron reactions in vitro, it has been argued that free radicals cannot be effectively scavenged under in vivo conditions by many nutritional antioxidants [21]. This argument is based on two assumptions. First, the most reactive free radical (i.e., hydroxyl radical) reacts with biological molecules (e.g., proteins) in the cell with the same rate constant as the rate of an antioxidant-radical reaction [21]. Thus, dietary antioxidants do not have an advantage over other cellular molecules in terms of antioxidant-oxidant reaction rates. Second, the cellular locations and limited concentrations of nutritional antioxidants are both drawbacks that limit the rate of radical removal from the cell. In this regard, Forman et al. contends that except for vitamin E and glutathione (GSH), the concentrations of nutritional antioxidants achieved in cells cannot overcome the reaction rate limitation to be an effective antioxidant in vivo [21]. Hence, these constraints underline the importance of having another line of defense in protection against radical mediated damage in cells. This additional line of defense against oxidative damage is a collection of antioxidant enzymes that work as a unit to lower ROS levels in organisms [22]. Importantly, cellular antioxidant enzymes are distributed across cellular compartments (i.e., cytosol, peroxisome, and mitochondria) to facilitate removal of oxidants at the site of production (Figure 2). These key cellular antioxidant enzymes are introduced in the next section. For a detailed discussion of antioxidant enzymes the reader is referred to comprehensive reviews on this topic [4,22]. 

### 3.1. Superoxide Dismutase (SOD)

As mentioned earlier, SOD is responsible for the dismutation of O_2_^•−^ radicals into H_2_O_2_ [1]. Three isoforms of SOD exist in mammals and all require a transition metal in the active site for enzymatic activity [23]. Two of these SOD isoforms are located within the cell (SOD1 and SOD2) whereas the third isoform (SOD3) is located within the extracellular space [24]. SOD1 requires copper and zinc as cofactors and is located in both the cytosol and the mitochondrial intermembrane space [24]. In contrast, SOD2 is located only within the mitochondrial matrix and requires manganese as a cofactor [24]. Similar to SOD1, SOD3 requires both copper and zinc as cofactors [25,26].

As introduced earlier, the O_2_^•−^ radical is the parent of all ROS in cells. Although O_2_^•−^ radicals are not highly reactive, these radicals can lead to the production of the highly reactive radicals (e.g., hydroxyl radicals and peroxynitrite) as well as the production of the oxidant, H_2_O_2_. Although H_2_O_2_ is not a radical, H_2_O_2_ is a ROS that can diffuse long distances and form more reactive species to promote oxidation. Hence, elimination of both the O_2_^•−^ radical and H_2_O_2_ are important in protecting cells against oxidation. Because of the importance of removing H_2_O_2_ from cells, several different antioxidant enzymes exist in various cellular compartments to achieve this goal (i.e., glutathione peroxidase, catalase, peroxiredoxins). A brief introduction of these enzymes follows.

### 3.2. Glutathione Peroxidase (GPX)

Eight isoforms of GPX exist and all GPX enzymes require selenium as a cofactor and use reducing equivalents from GSH to reduce both H_2_O_2_ and organic hydroperoxides to form water or alcohol [4,27,28]. The expression of multiple isoforms of GPX is logical because of the different locations of the isoforms [27]. For instance, GPX1 is found in both mitochondria and the cytosol of muscle fibers [27]. While GPX2 is located exclusively within the cytosol, GPX3 is located within both the cytosol and extracellular space [27]. Further, GPX4 is also located in the mitochondria and is specialized in reducing lipid peroxide; notably, GPX4 has been shown to play a key role in the regulation of ferroptosis [29]. This compartmentalization of the GPX family of enzymes allows these enzymes to reduce peroxides across multiple locations within cells. 

### 3.3. Catalase (CAT)

Although the term “catalase” was not coined until 1900, the origin of CAT, an enzyme that degrades H_2_O_2_ to water and oxygen, dates back to the 19^th^ century [30]. Specifically, it was recognized in the 1800′s that cells contained substances capable of eliminating H_2_O_2_ and it was later discovered that this substance was CAT; this remarkable history makes CAT the first discovered antioxidant enzyme [30].

Functionally, CAT requires iron as a cofactor and has one of the highest turnover rates of all enzymes. Moreover, CAT does not require reducing equivalents to eliminate H_2_O_2_. However, unlike GPX, CAT cannot reduce organic hydroperoxides. To remove H_2_O_2_ from different regions of the cell, CAT exists in several cellular compartments including peroxisomes, the cytosol, and mitochondria. However, the precise location of catalase within the mitochondrion remains a topic of debate. Notably, the Km of GPX and CAT for H_2_O_2_ differ (0.1mM and 1.0 mM, respectively) such that each enzyme catalyzes the removal of H_2_O_2_ efficiently within its in vivo substrate concentration range [31]. 

### 3.4. Peroxiredoxins (PRDXs)

For several decades, CAT and GPXs were considered the major peroxide reducing enzymes in cells [32]. However, the discovery of a third group of abundant peroxidases in 1994 triggered a paradigm shift that led to the current consensus that a family of enzymes termed peroxiredoxins (PRDXs) play a key role in the removal of peroxides in cells [33]. These enzymes differ from CAT and GPX in several ways including the fact that they do not require a cofactor and use cysteine residues for catalysis [32,34]. The consensus belief that PRDXs play a dominant role in removal of peroxides from cells grew from the discovery that these enzymes are efficient catalysts, having second-order rate constants comparable to both CAT and GPX [32,34]. More importantly, PRDXs are expressed in cells at higher levels compared to both CAT and GPX and notably, evidence indicates that in human cells, >99% of peroxide in the cytosol and >90% of peroxide in the mitochondria are eliminated by PRDX enzymes [32]. Hence, peroxide removal from cells occurs via the interaction of three enzymes (i.e., CAT, GPX, PRDXs) with the PRDX family of enzymes playing the dominant role. 

Six isoforms of PRDX exist (PRDX1-6) and each of these enzymes reduce H_2_O_2_, alkyl peroxides, and peroxynitrite using electrons provided by thioredoxin. Similar to other cellular antioxidant enzymes, the various isoforms of PRDX are located in varying cellular locations including the cytosol, nucleus, mitochondria, and peroxisomes [34]. For example, human PRDX5 is located in both the cytosol and mitochondria. Moreover, growing evidence indicates that PRDX play a role in cell signaling events in skeletal muscle fibers and other cells [34,35]. 

### 3.5. Thioredoxins (Trxs)

Thioredoxins (Trxs) are inducible antioxidant proteins expressed in all cell types. Briefly, Trxs are low-molecular weight proteins that reduce numerous target enzymes including PRDXs [36]. The Trx antioxidant system contains two primary components: (1) Trx; and (2) Trx reductase (TrxR). Three isoforms of TrxR exist with TrxR1 being located in the cytosol whereas TrxR2 is found within the mitochondria [36]. Another TrxR isoform (TrxR3) is located within the testes [37]. Trx reduces cellular enzymes by transferring two electrons to the target protein, resulting in reduction of the substrate. The oxidized form of Trx is then reduced by TrxR using electrons from NADPH; this allows Trx to continue as a redox modulator [36]. 

As an antioxidant, Trx serves several physiological functions in protecting cells against oxidative damage. First, Trx appears to exert most of its antioxidant activity by supplying electrons to methionine sulfoxide reductases (i.e., PRDXs) and its action as a protein disulfide reductase [37]. The protein disulfide reductase activity of Trx is important in the regeneration of oxidized proteins [37]. Further, Trx can directly target antioxidant gene expression by regulating the DNA binding activity of several transcription factors including nuclear factor kappa B (NF-kB), AP-1, and p53 [37].

Exercise training increases the abundance/activities of key antioxidant enzymes in skeletal muscles and the next sections discuss the supporting evidence. Specifically, we discuss the available evidence that endurance, high-intensity interval, and resistance exercise training all increase antioxidants in skeletal muscles. Note that preclinical studies and studies in human subjects are discussed in separate sections. 

## 4. Endurance Exercise Training-Induced Changes in Muscle Antioxidant Enzymes 

This section discusses the effects of endurance exercise training (i.e., continuous exercise at submaximal intensity) at varying exercise intensities (e.g., 50–75% VO_2_ max) on skeletal muscle antioxidant enzymes in both preclinical and human studies. We begin with a discussion of the data from preclinical studies investigating the impact of endurance exercise on antioxidant enzyme activity in skeletal muscles.

### 4.1. Preclinical Studies Reveal That Endurance Training Increases the Activity of Key Antioxidant Enzymes in Skeletal Muscles

The first study demonstrating that endurance exercise training increases muscle antioxidant enzyme activity appeared in 1983 [3]. In the decades following this groundbreaking report, numerous studies have provided details about the impact of endurance training on the activity of antioxidant enzymes in skeletal muscle. We begin with a summary of investigations into the effect of endurance exercise training on the activities of SOD1 and SOD2 in locomotor muscles. Although some studies report that endurance training does not increase SOD1 or SOD2 in trained limb muscles [38,39,40,41], many studies agree that endurance exercise training increases the activities of both SOD1 and SOD2 in skeletal muscles [3,42,43,44,45,46,47,48,49,50,51,52]. Interestingly, the magnitude of endurance training-induced increases in muscle SOD1 and SOD2 activity across studies ranges between 20–112%. The explanation for the variance across research findings is not clear but methodological diversity in the assay of SOD activities together with differences in exercise training protocols are possible contributors to this lack of accord. For example, 10-fold variances exist in the sensitivity between methods used to assay SOD activity [53]. 

Hence, assays of SOD activity with low sensitivity would not detect small-to-moderate differences in SOD activity between tissues. It is also possible that the failure of some studies to report exercise-induced increases in SOD activity in muscles is due to a low intensity and/or short duration of the exercise training sessions [51]. Indeed, studies exercising animals at higher relative intensities (e.g., >50% VO_2_ max) and/or longer durations (>30 min/day) typically report significant increases in the abundance/activity of both SOD1 and SOD2 [4]. Figure 3 illustrates this point and shows that total SOD activity (SOD1 + SOD2) increases in the rat gastrocnemius muscle as a function of both exercise intensity and duration [51]. 

Although the effect that endurance exercise training has on total SOD protein/activity in skeletal muscle has received significant research attention, investigations into the impact that endurance exercise has on specific SOD isoforms in muscle has been studied sparsely. While one study concludes that endurance exercise training increases SOD2 protein content/activity in skeletal muscles without increasing SOD1 abundance [42], other studies conclude that endurance exercise training increases the protein expression of both SOD1 and SOD2 in trained rat skeletal muscles [49,54]. The explanation for these divergent findings is not clear but could be due to differences in the muscles selected for study (i.e., different exercise recruitment patterns between the limb muscles investigated).

Similar to SOD, variations exist in the literature about the impact of endurance exercise training on total GPX activity in skeletal muscles. Nonetheless, most preclinical studies conclude that endurance exercise training results in significant increases (e.g., +20–177%) in total GPX activity in skeletal muscles (reviewed in [6]). Similar to the exercise-induced increases in SOD, an exercise dose–response relationship exists between the exercise intensity/daily training duration and the magnitude of increases in total GPX activity in skeletal muscles (Figure 4) [51].

Currently, most published studies conclude that endurance exercise training increases total GPX activity in the trained muscles. However, limited information exists about the impact of endurance exercise on the expression of the specific GPX isoforms within skeletal muscle fibers. 

Numerous studies have also investigated the impact of endurance exercise training on the expression/activity of catalase in rodent skeletal muscles. Unfortunately, no clear consensus exists as to whether exercise training increases catalase activity in skeletal muscles. Indeed, studies have reported that endurance exercise training increases, decreases, or does not change catalase activity in skeletal muscles [4]. The explanation for these divergent findings remains unclear but may be the result of challenges in the assay of catalase activity; see reference [4] for a detailed discussion of these problems. 

Finally, while it is established that both PRDX and TRX are key antioxidants in cells, the impact of endurance exercise training on the expression/activity of these proteins in skeletal muscles of animals remains unknown. Clearly, this is an important topic for future research. 

### 4.2. Endurance Exercise Training Increases Key Antioxidant Enzymes in Human Skeletal Muscles

Compared to the large number of preclinical studies investigating the impact of endurance training on the activity of antioxidant enzymes in skeletal muscles, human studies on this topic are limited. The obvious explanation for the small number of human studies is the invasive nature of obtaining skeletal muscle biopsies. Nonetheless, several clinical studies exist and collectively, these studies confirm that endurance exercise training increases the activities of key antioxidant enzymes in human skeletal muscles. Although an early study concluded that short-term aerobic exercise training does not increase antioxidant enzyme activities in human skeletal muscles [55], subsequent studies have consistently demonstrated that weeks-to-months of endurance exercise increases antioxidant enzyme activity in the trained muscles. For example, six months of endurance exercise training increases the activities of total SOD, CAT, and total GPX in skeletal muscle by 31%, 57%, and 51%, respectively [56]. Similarly, another study concluded that moderate-intensity endurance exercise training increases the expression of SOD2, GPX1, and PRDX5 in skeletal muscles by 66%, 62%, and 37%, respectively [57]. Endurance exercise has also been reported to increase both SOD and CAT in skeletal muscles of obese sedentary men [58]. Finally, a retrospective study concluded that older adults engaged in regular bouts of moderate-to-vigorous physical activity have higher levels of total SOD, total GPX, and CAT activity in locomotor skeletal muscles compared to inactive individuals [59]. Similarly, another study investigating the impact of life-long exercise on muscle antioxidants in humans concluded that endurance exercise increases the abundance of both SOD2 and PRDX5 protein in the trained locomotor muscles [60]. Collectively, these clinical studies are generally consistent with the preclinical studies indicating that endurance exercise training increases the activities of key antioxidant enzymes in trained skeletal muscles of humans. 

## 5. High-Intensity Interval Training-Induced Changes in Skeletal Muscle Antioxidant Enzymes

High-intensity interval exercise training (HIIT) involves performing short bouts of vigorous exercise followed by a short recovery period. These short bouts of exercise can last between 10 s to a few minutes with exercise intensities ranging from 80–150% of VO_2_ max. In this segment we discuss the impact of high-intensity interval training on the activities of antioxidant enzymes in skeletal muscles. 

### 5.1. Preclinical Studies

Two independent preclinical studies conclude that HIIT results in selective increases in antioxidant enzyme activity in rat locomotor muscles. Interestingly, these studies used markedly different training protocols but both studies demonstrated that HIIT increases total GPX activity in trained skeletal muscles. For example, Atalay et al. trained animals for six weeks using a sprint HIIT protocol (i.e., short bouts of maximal exercise); this mode of HIIT did not elevate total SOD activity in muscle fibers but significantly increased the activities of both GPX (total) and glutathione reductase in the trained locomotor muscles [61]. Using a markedly different HIIT approach (5-minute intervals at ~80–95% VO_2_ max), Criswell et al. concluded that this mode of HIIT increased both total GPX and total SOD activity in limb muscles [62]. Together, these results reveal that different intensities of HIIT can selectively increase the activity of antioxidant enzymes in the trained skeletal muscles. Note, however, that the specific muscle antioxidant enzymes impacted by HIIT appear to be influenced by the intensity and duration of the interval training. 

### 5.2. High-Intensity Interval Training and Antioxidant Enzymes in Human Skeletal Muscles

To date, only one human study has investigated the impact of HIIT on skeletal muscle antioxidant enzyme activity. Briefly, subjects performed 10 s of high-intensity exercise followed by 50 s of rest; this training interval was repeated 15 times per day for seven weeks. The results revealed that seven weeks of exercise training increased total GPX activity in skeletal muscles by 27% [63]. In contrast, this mode of HIIT did not increase the expression of SOD in skeletal muscles [64]. Nonetheless, this study demonstrates that, by increasing the abundance of GPX, HIIT improves the ability of skeletal muscle fibers to eliminate hydrogen peroxide and other organic hydroperoxides.

## 6. Resistance Training-Induced Increases in Muscle Antioxidant Enzymes

In comparison to endurance exercise training studies, investigations of the impact of resistance exercise training on antioxidant enzyme activity in skeletal muscles are limited. The next sections highlight our existing knowledge about the impact of resistance exercise on skeletal muscle antioxidant enzyme activity. 

### 6.1. Preclinical Studies Suggest That Resistance Training Increases Skeletal Muscle Antioxidants

To date, only a few preclinical studies have investigated the impact of resistance training on the activities of antioxidant enzymes [65,66,67]. This paucity of studies is likely due to the complexity of conducting rigorous resistance training protocols using animal models. For example, only a few studies have confirmed that skeletal muscle hypertrophy is achievable following resistance training in rats. Nonetheless, the available evidence indicates that resistance exercise training in rats increases the activities/abundance of several antioxidant enzymes in skeletal muscles including total SOD, SOD1, and total GPX [65,67]. In contrast, resistance training does not appear to increase the activity/abundance of CAT in skeletal muscles [65,67]. Currently, no preclinical studies have investigated the impact of resistance exercise training on the expression of PRDX or Trx in skeletal muscles. 

### 6.2. Resistance Exercise Training Increases Antioxidant Enzyme Activity in Humans 

Studies investigating the impact of resistance exercise training on antioxidant enzyme activity in human skeletal muscles corroborate preclinical findings and show that resistance exercise training increases the activities of several antioxidant enzymes. Indeed, studies in type II diabetics, middle-age men, and older adults all conclude that resistance exercise training increases the activities of several key antioxidant enzymes in skeletal muscles [57,68,69,70,71]. Furthermore, an acute bout of resistance training increases muscle mRNA levels of CAT, SOD1, SOD2, and GPX. Although differences often exist between mRNA levels and protein abundance in muscle, several studies conclude that resistance exercise increases the activities of SOD1, SOD2, CAT, and GPX (total) in human skeletal muscle [57,69,70,71]. Further, evidence also exists that resistance training increases the activities of glutathione reductase, PRDX5, and TrxR1 in human muscle [57,68,70]. Together, these studies confirm that, similar to endurance and interval training, resistance exercise training improves the antioxidant capacity in human skeletal muscle. The question now becomes, “Does this exercise-induced increase in muscle antioxidant enzyme activity protect skeletal muscle fibers against ROS-mediated oxidative damage?”.

## 7. Exercise-Induced Improvements in Muscle Antioxidant Enzyme Capacity Protects against ROS-Mediated Oxidative Damage

The previous sections (i.e., Section 4, Section 5 and Section 6) highlight the available evidence that exercise training increases antioxidant enzymes in trained skeletal muscles of humans and other animals. Importantly, both preclinical and human studies reveal that although the exercise-induced increases in muscle antioxidant enzyme activity is often small, this increased antioxidant capacity is sufficient to provide protection against several sources of oxidative stress. For example, preclinical studies disclose that endurance exercise training increases the antioxidant capacity of diaphragm muscle [72,73] and protects against contraction-induced oxidative stress (e.g., reduction in lipid peroxidation within muscle) [72,73,74,75,76]. Further, as few as five consecutive days of endurance exercise increases muscle antioxidant enzyme activity (i.e., SOD and CAT) and protects against exercise-induced oxidative damage [76]. Notably, this exercise-induced increase in muscle antioxidants is associated with improved muscular endurance [76]. Nonetheless, although contraction-induced ROS production can contribute to muscle fatigue [77,78,79,80], it is uncertain as to whether this improvement in fatigue resistance with training is primarily due to the training-induced increase in muscle antioxidants. 

Preclinical studies also show that endurance exercise training protects skeletal muscle against oxidants resulting from sources other than muscular exercise. For example, endurance exercise training protects diaphragm muscle from cigarette smoke-induced oxidative damage/wasting [81]. Similarly, endurance exercise training also protects the diaphragm from ventilator-induced oxidative damage and fiber atrophy [82,83,84]. More specifically, recent studies confirm that the exercise-mediated protection against ventilator-induced oxidative stress/atrophy is due, in part, to exercise-induced increases in SOD2 [82]. A final example of exercise training-induced protection against oxidative stress is provided by studies showing that endurance exercise training protects against doxorubicin-mediated muscle wasting [85]. Although doxorubicin is a potent antitumor agent used in cancer treatment, it is also myotoxic. This doxorubicin-induced myotoxicity is due, in part, to the generation of ROS in muscle fibers; this doxorubicin-induced oxidative stress also results in the activation of muscle proteases resulting in fiber atrophy. Therefore, it is postulated that exercise-induced improvements in muscle antioxidant capacity is the mechanism responsible for exercise-induced protection against doxorubicin-induced muscle wasting [85].

Similar to the aforementioned preclinical studies, several human studies confirm that exercise training attenuates exercise-induced oxidative stress. For example, resistance exercise training lowers endurance exercise-induced oxidative stress in both overweight and obese older adults [86]. Similarly, resistance training also attenuates exercise-induced lipid peroxidation in non-obese adults [87]. Further, endurance exercise training has also been shown to reduce exercise-induced oxidative stress in non-obese older men [88]. Together, these preclinical and clinical studies confirm that the exercise training-induced increases in muscle antioxidant capacity is sufficient to protect against subsequent acute bouts of exercise-induced oxidative stress. 

## 8. Conclusions and Future Directions

The discovery that muscular exercise results in oxidative stress, due in part to ROS production by the active skeletal muscles, was first reported over 50 years ago. Although debate continues about the primary sites of ROS production in contracting muscle fibers, growing evidence indicates that NOX is an important site of ROS production in contracting muscles. Moreover, evidence also suggests that both PLA2 and xanthine contribute to the increase in muscle ROS production during exercise. 

Cells are protected against oxidant damage by two layers of antioxidant protection. Antioxidant enzymes are a key part of this protection; these enzymatic antioxidants are distributed in various cellular compartments to facilitate removal of ROS at the site of production. Abundant evidence in both preclinical and clinical studies indicates that endurance exercise training increases the activities of SOD (total), SOD1, and SOD2 in the trained skeletal muscles. Similarly, endurance exercise training increases the activity of total GPX in skeletal muscles of humans and other animals. Finally, preclinical studies fail to provide a consensus as to whether endurance exercise training increases the abundance of CAT in the trained muscle fibers. In contrast, clinical studies suggest that endurance exercise training is effective in increasing the abundance of CAT in human skeletal muscles (Figure 5). Together, these preclinical and clinical studies confirm that endurance exercise training increases muscle antioxidant capacity and provides protection against subsequent bouts of exercise-induced oxidative stress. 

Compared to studies investigating the impact of endurance training on skeletal muscle antioxidant capacity, limited studies have examined the influence of HIIT on antioxidant enzymes. Nonetheless, the available evidence indicates that HIIT increases both SOD and GPX activity in skeletal muscles of exercise trained rodents. Similarly, a human study confirms that HIIT increases total GPX activity in muscle. However, this study failed to report an increase in SOD or CAT in the trained muscles (Figure 5).

Again, compared to the large number of investigations into the impact of endurance training on muscle antioxidants, studies examining the effects of resistance training on skeletal muscle antioxidant enzymes are limited. Nonetheless, preclinical studies indicate that resistance training increases the abundance/activity of total SOD, SOD1, and total GPX in rodent skeletal muscles. Clinical studies confirm these preclinical findings and demonstrate that resistance exercise training increases the activities of SOD1, SOD2, GPX, and CAT in human skeletal muscles (Figure 5). 

Although much progress has been made in understanding the impact that several modes of exercise training have on muscle antioxidant capacity, many unanswered questions remain. For example, limited evidence exists regarding the effects that various types of HIIT have on antioxidant enzyme abundance/activity in skeletal muscles. Further, few studies have investigated the influence of exercise training on the levels of TRX isoforms and the six isoforms of PRDX in trained muscles. Given the importance that PRDX plays in the removal of peroxides in cells, these future studies are essential to gain a complete understanding of exercise-induced improvements in muscle antioxidant capacity. Clearly, there is much more to learn about this exciting topic.

## Figures and Tables

**Figure 1 antioxidants-12-00039-f001:**
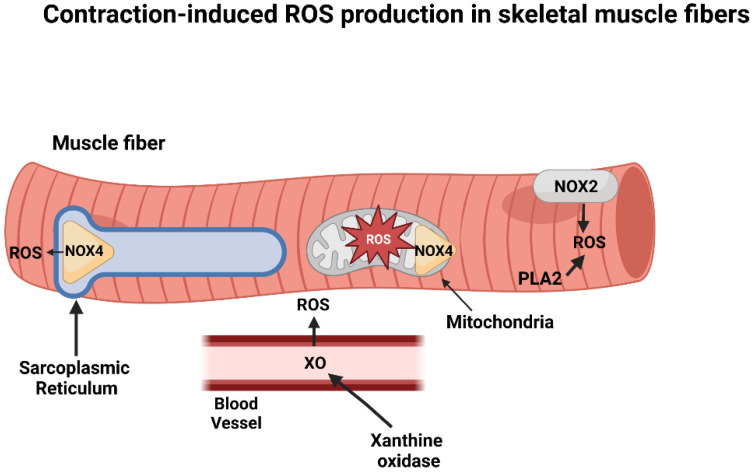
Potential sites of ROS production in skeletal muscle fibers during contractile activity. See text for details. Key: NOX2 = NAD(P)H oxidase 2; NOX4 = NAD(P)H oxidase 4; PLA2 = phospholipase A2; ROS = reactive oxygen species. Figure created by BioRender.com.

**Figure 2 antioxidants-12-00039-f002:**
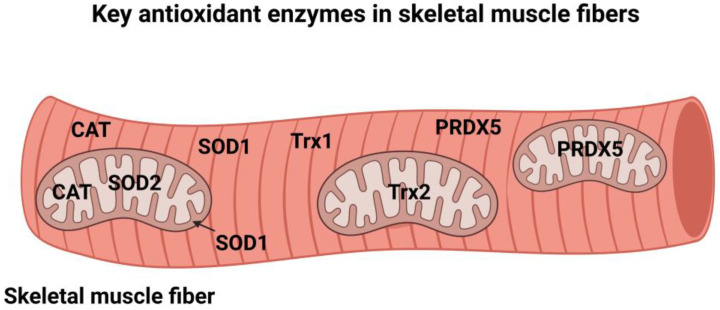
Cellular locations of key antioxidant enzymes in skeletal muscle fibers. See text for specific details. Key: SOD1 = superoxide dismutase 1; SOD2 = superoxide dismutase 2; GPX = glutathione peroxidase; CAT = catalase; Trx1 = Thioredoxin 1; Trx2 = Thioredoxin 2; PRDX5 = Peroxiredoxin 5. Figure created by BioRender.com.

**Figure 3 antioxidants-12-00039-f003:**
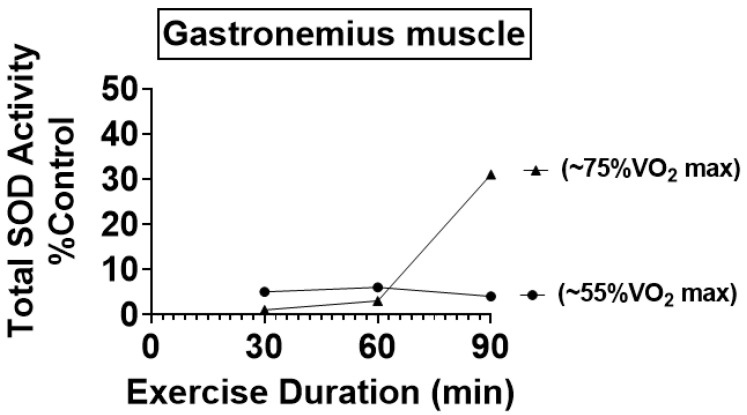
Illustration of the exercise dose–response increases in total SOD activity (SOD + SOD2) in the rat soleus muscle following 10 weeks of endurance exercise training. Data are from reference [51].

**Figure 4 antioxidants-12-00039-f004:**
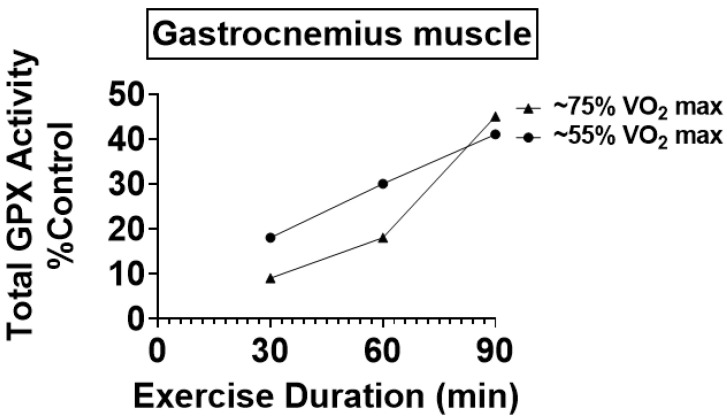
Illustration of the exercise dose–response increases in total GPX activity in the rat soleus muscle following 10 weeks of endurance exercise training. Data are from reference [51].

**Figure 5 antioxidants-12-00039-f005:**
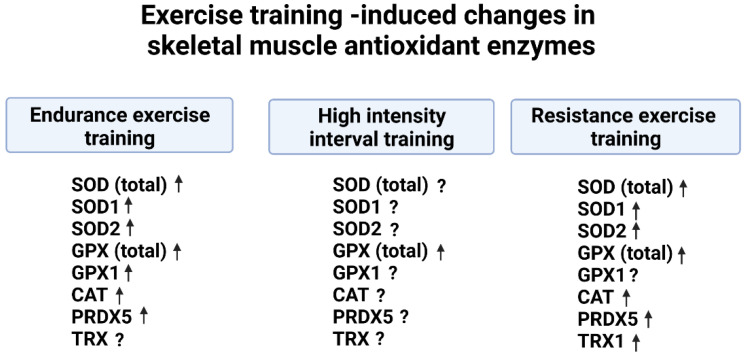
Summary of the effects of exercise training on the abundance/activity of antioxidant enzymes in skeletal muscle. SOD (total) = superoxide dismutase 1 + superoxide dismutase 2; SOD1 = superoxide dismutase 1; SOD2 = superoxide dismutase 2; GPX (total) = all glutathione peroxidase isoforms in fiber; GPX1 = glutathione peroxidase 1; CAT = catalase; Trx1 = Thioredoxin 1; PRDX5 = Peroxiredoxin 5 = PRDX5. Arrows signify exercise-induced increases in antioxidant activity. Question marks (?) indicate that it remains unknown as to whether exercise increases the activity of select antioxidants. Figure created by BioRender.com.

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
