# Peer review of "Exercise Training and Skeletal Muscle Antioxidant Enzymes: An Update"

_antioxidants, 2022, doi:10.3390/antiox12010039_

Round 1
Reviewer 1 Report
Thank you for the opportunity to review this manuscript.
This review contributes to the current body of literature on the impact of different types of exercise and intensities, and time duration on antioxidant enzyme activities, protein content, and gene expression in the skeletal muscle. The authors also report that exercise may induce the regulation of cellular antioxidants and alleviate contraction-induced oxidative damage in the skeletal muscles. In addition, physical activity has the potential to delay muscle fatigue and improve exercise performance.
The authors have prepared an acceptable manuscript. Moreover, from the clinical perspective data presented in this review is very important, especially in the area of the physiology of exercise. However, I have some concerns with the manuscript exist, which may confuse the readers of Antioxidants. There are some suggestions and recommendations listed below, to clarify and strengthen this interesting work.
Major:
Could the authors cite Fig. 2 in the text of the manuscript, please?
Line 176,
Could the authors cite Fig. 2 in the text of the manuscript, please?
I would kindly like to ask you to change the focus to catalase localization in mitochondria. Catalase activity in the muscle is very low as compared to the liver, less than 1.5%. Therefore, mitochondria isolated from the skeletal muscle possess very low catalase activity, so it is possible to assume that it does not play an important role as an antioxidant enzyme. Moreover, it raises another concern about mitochondrial subpopulations, both across fibre types (fast-glycolytic and slow-oxidative twitch) as well as within the cell (subsarcolemmal and intermyofibrillar). Catalase activity is 3-4 fold higher in slow as compared to fast twitch fibre. In addition, showing that catalase is located in the matrix of muscle mitochondria may be confusing. Does it mean that there is the same amount of red and white muscle? Please change the localization of catalase in Fig. 2 or add more details to the manuscript related to muscle fiber specific. Furthermore, it would be good to add the location of catalase in the mitochondrial matrix, which is not fully confirmed according to various authors.
Additionally, in the same way, should be presented SOD1 in the mitochondrial membrane space.
Minor:
There are some typo errors such as pre-clinical; preclinical, Fig. 5; Figure 4
Line 371….. training increases muscle mRNA levels of CAT, SOD1, SOD2, and GPX. Although differences often exist between mRNA levels and protein abundance….
Instead, of levels, it is better to use expression. Please check the whole manuscript and make appropriate changes.
Author Response
Responses to reviewer 1
The authors thank the reviewer for their time invested in the review of our manuscript. Indeed, we are grateful for your suggested changes to our review and have responded to each of your comments in a point-by-point fashion. The following changes have been made in the revised manuscript:
Reviewer comment 1: Could the authors cite Fig. 2 in the text of the manuscript, please?
Author(s) response: We apologize for the failure to cite Figure 2 in the text of the manuscript. This error has been corrected in revised manuscript.
Reviewer comment 2: I would kindly like to ask you to change the focus to catalase localization in mitochondria. Catalase activity in the muscle is very low as compared to the liver, less than 1.5%. Therefore, mitochondria isolated from the skeletal muscle possess very low catalase activity, so it is possible to assume that it does not play an important role as an antioxidant enzyme. Moreover, it raises another concern about mitochondrial subpopulations, both across fibre types (fast-glycolytic and slow-oxidative twitch) as well as within the cell (subsarcolemmal and intermyofibrillar). Catalase activity is 3-4 fold higher in slow as compared to fast twitch fibre. In addition, showing that catalase is located in the matrix of muscle mitochondria may be confusing. Does it mean that there is the same amount of red and white muscle? Please change the localization of catalase in Fig. 2 or add more details to the manuscript related to muscle fiber specific. Furthermore, it would be good to add the location of catalase in the mitochondrial matrix, which is not fully confirmed according to various authors.
Author(s) response: Thank you for these comments. As suggested, the revised text indicates that the precise location of catalase in the mitochondrion remains a topic of debate.
Reviewer comment 3: Additionally, in the same way, should be presented SOD1 in the mitochondrial membrane space.
Author(s) response: The fact that SOD1 is located in both the cytosol and mitochondrial intermembrane space is now recognized in the revised Figure 2. We apologize for the omission in the original figure.
Reviewer comment 4: There are some typo errors such as pre-clinical; preclinical, Fig. 5; Figure 4
Author(s) response: The four typos for “pre-clinical” have been corrected in the revised text.
Reviewer comment 5: Line 371….. training increases muscle mRNA levels of CAT, SOD1, SOD2, and GPX. Although differences often exist between mRNA levels and protein abundance….Instead, of levels, it is better to use expression. Please check the whole manuscript and make appropriate changes.
Author(s) response: Thank you for this comment. Where appropriate, we have substituted the word “expression” for “abundance” in the revised text.
Reviewer 2 Report
This is an interesting and a well-written review. The authors nicely summarize current knowledge on antioxidant enzyme activity and abundance after three different types of exercise training, namely, endurance training, HIIT and resistance training. They also distinguish preclinical from human studies in order to more appropriately cover the topic.
Based on their analysis, the authors report:
· A large number of studies have been conducted about the impact of endurance training on redox enzymes. Collectively, both preclinical and clinical studies confirm that endurance exercise training increases muscle antioxidant capacity and provides protection against subsequent bouts of exercise-induced oxidative stress.
· Limited studies have examined the influence of HIIT on the aforementioned enzymes. The available evidence indicates that HIIT increases both SOD and GPX activity in skeletal muscles of exercise trained rodents. Likewise, a human study confirms that HIIT increases total GPX activity in muscle, but failed to report an increase in SOD or CAT in the trained muscles.
· Regarding the impact of resistance training on skeletal muscle antioxidant enzymes, preclinical studies indicate that resistance training increases the abundance/activity of skeletal muscle total SOD, SOD1, and total GPX in rodents. Clinical studies confirm these preclinical findings in humans.
Please find next some minor comments that may be of interest:
· Regarding Figure 1: apart from XO, blood cells could perhaps be also a source of ROS in blood vessels that in some cases (depending on features like cell membrane permeability, extracellular space volume/area ratio, intracellular H2O2 clearance capacity, mode of extracellular H2O2 production and extracellular fluid fluxes) can reach neighboring skeletal muscle cells? A recent study provided some interesting data (PubMedID: 36335761).
· Section 2 ‘Cellular antioxidant enzymes’, lines 119-121: my apologies if I misunderstood something about this statement, but…if only vitamin E and GSH of the nutritional antioxidants can overcome the reaction rate limitation to be effective antioxidants in vivo, this means that, for example, vitamin C supplementation should not affect oxidative stress levels. However, this is not the case. Please clarify.
· Section 2 ‘Cellular antioxidant enzymes’: given the superiority of antioxidant enzymes against ROS based on the kinetic rates of the reactions (PubMedID: 30148624), why do the authors label these enzymes as ‘second’ line of defense and not the first one? Please clarify.
· Regarding resistance training: the authors provide the preclinical studies [65-67] as resistance training, but #66 includes running on a treadmill without inclination or declination. Please clarify if this is indeed resistance training.
· At the end of section 7, the authors additionally provide some human studies which confirm that exercise training attenuates exercise-induced oxidative stress, with the measurements being performed in blood cells or plasma. There is also a recent human study that corroborates these findings (PubMedID: 28544643, Table 3 and Figures 4 and 5)
Author Response
Response to reviewer 2
The authors thank the reviewer for their time invested in the review of our manuscript. Indeed, we are grateful for your suggested changes to our review and have responded to each of your comments in a point-by-point fashion. The following changes (reflected in red font) have been made in the revised manuscript:
Reviewer comment 1: Regarding Figure 1: apart from XO, blood cells could perhaps be also a source of ROS in blood vessels that in some cases (depending on features like cell membrane permeability, extracellular space volume/area ratio, intracellular H2O2 clearance capacity, mode of extracellular H2O2 production and extracellular fluid fluxes) can reach neighboring skeletal muscle cells? A recent study provided some interesting data (PubMedID: 36335761).
Author(s) response: Thank you for this comment and providing a link to this interesting paper. While the authors appreciate that blood cells can produce ROS, to our knowledge, evidence indicating that blood cells are a major source of ROS production during exercise does not exist. Moreover, our discussion of exercise-induced ROS production is limited to sources of ROS in contracting skeletal muscles (lines 61-105).
Reviewer comment 2: Section 2 ‘Cellular antioxidant enzymes’, lines 119-121: my apologies if I misunderstood something about this statement, but…if only vitamin E and GSH of the nutritional antioxidants can overcome the reaction rate limitation to be effective antioxidants in vivo, this means that, for example, vitamin C supplementation should not affect oxidative stress levels. However, this is not the case. Please clarify.
Author(s) response: Thank you for this comment. Please note that the statement (lines 120-123) about vitamin E and GSH is a direct quote from Forman et al. (21) and is not a statement original to the authors. Whether Forman et al. is correct on this account is subject to debate. However, we believe that Forman’s point is that the concentration of C in cells resulting from a balanced diet is not sufficient to be a highly effective antioxidant.
Reviewer comment 3: Section 2 ‘Cellular antioxidant enzymes’: given the superiority of antioxidant enzymes against ROS based on the kinetic rates of the reactions (PubMedID: 30148624), why do the authors label these enzymes as ‘second’ line of defense and not the first one? Please clarify.
Author(s) response: Thank you for this comment. We agree with your concern about the labeling of “first and second” lines of antioxidant defense in the cells. Therefore, this section of text has been revised to eliminate these labels.
Reviewer comment 4: Regarding resistance training: the authors provide the preclinical studies [65-67] as resistance training, but #66 includes running on a treadmill without inclination or declination. Please clarify if this is indeed resistance training.
Author(s) response: Thank you for bringing this point to our attention. Reference #66 was incorrectly cited in line 364. This error (reference removed) has been corrected in the revised manuscript. We are grateful for your careful review of our report.
Reviewer comment 5: At the end of section 7, the authors additionally provide some human studies which confirm that exercise training attenuates exercise-induced oxidative stress, with the measurements being performed in blood cells or plasma. There is also a recent human study that corroborates these findings (PubMedID: 28544643, Table 3 and Figures 4 and 5)
Author(s) response: Thank you for bringing this paper to our attention.